# Rare Prenyllipids in Wild St. John’s Wort During Three Harvest Seasons

**DOI:** 10.3390/molecules30040901

**Published:** 2025-02-15

**Authors:** Paweł Górnaś, Aleksander Siger

**Affiliations:** 1Institute of Horticulture, Graudu 1, LV-3701 Dobele, Latvia; 2Department of Food Biochemistry and Analysis, Poznan University of Life Sciences, Wojska Polskiego 28, 60-637 Poznań, Poland; aleksander.siger@up.poznan.pl

**Keywords:** RP-HPLC-FLD, hypericaceae, medical plant, aerial part, tocol, lipophilic bioactive compound

## Abstract

St. John’s wort (*Hypericum perforatum*) is a medicinal plant known for its bioactive compounds, including tocopherols and tocotrienols, which possess antioxidant and anti-inflammatory properties. These compounds play vital roles in the plant’s metabolism and have potential applications in the cosmetic and pharmaceutical industries. However, the content of these compounds in different anatomical parts of the plant, as well as the influence of environmental factors, such as the year of collection, remain underexplored. This study examined the content of tocochromanols in *H. perforatum* leaves, flowers, and flower buds, collected in Poland during the years 2022–2024. The results revealed that tocopherols predominantly accumulated in the leaves, while tocotrienols were more abundant in the flowers and flower buds. The year of collection had a significant effect on tocopherol levels, while tocotrienol content showed lower sensitivity to environmental fluctuations, indicating their higher stability. St. John’s wort can be considered a valuable source of biologically active compounds, especially tocotrienols, which exhibit higher stability and less susceptibility to environmental variability. The results underline the importance of considering both the plant’s anatomical parts and the year of collection when aiming to maximize the production of bioactive compounds.

## 1. Introduction

The growing interest in alternative medicine and natural health products is a significant growth factor for the market of herbal products and medicines. In developing countries, approximately 80% of the population relies on medicinal and aromatic plants, such as St. John’s wort, to meet their basic healthcare needs [1]. Consumers are increasingly turning to herbal remedies as a safer and more natural alternative to synthetic pharmaceuticals. The herbal medicine market size in the United States has been estimated at approximately USD 7.45 billion [2]. With a trade volume exceeding USD 6 million in this market, *Hypericum* species hold a very important position. The global market size for *Hypericum perforatum* extract was valued at approximately USD 200 million in 2023, and it is projected to reach around USD 350 million by 2032 (https://dataintelo.com/report/global-hypericum-perforatum-extract-market, accessed on 6 February 2025). The genus *Hypericum* includes approximately 500 species of herbs, shrubs, and trees distributed across all continents of the world except Antarctica [3]. However, only *H. perforatum* is officially recognized as an herbal medicine in several pharmacopoeias and is used clinically as an antidepressant [4]. In *Hypericum* spp., over 100 different metabolites belonging to different groups of compounds have been identified [5]. The most investigated species in the genus *Hypericum* is *H. perforatum* [6,7,8]. Beneficial therapeutic effects of plant extracts from *Hypericum* spp. include remarkable wound healing, antimicrobial, antidepressant-like, and antinociceptive effects, as well as cytotoxic and anti-inflammatory activity. Antitumor potential for cancer-related inflammation of plant extracts from *Hypericum* spp. has been associated with the presence of bioactive molecules [9]. The study by Ji et al. [4] demonstrated differences in the chemical composition among five *Hypericum* species. The three species exhibiting the highest cytotoxic activity against cancer cell lines were *H. attenuatum*, *H. erectum*, and *H. perforatum*. These species were characterized by high concentrations of petiolin A, prolificin A, and hypercohin G, respectively. *H. faberi* and *H. perforatum* contained the highest concentrations of anti-inflammatory compounds, such as pseudohypericin, quercetin, and chlorogenic acid. The study also revealed that *H. perforatum* and *H. erectum* showed stronger anti-plasmodial activity against *Plasmodium falciparum* 3D7, attributed to hyperforin and xanthones.

While much of the research on *H. perforatum* has concentrated on the biological potential of hydrophilic compounds [10,11,12,13], there remains a limited understanding of the presence and significance of lipophilic bioactive constituents, such as tocopherols (Ts) and tocotrienols (T3s) [14,15,16,17]. Furthermore, the variability in the levels of these lipophilic compounds across different studies indicates inconsistencies that warrant further investigation. The identification of tocopherol and tocotrienol homologues in various medicinal plants, including St. John’s wort, was first accomplished using electrospray ionization coupled with liquid chromatography-tandem mass spectrometry (ESI(+)-LC-MS/MS) over a decade ago (in 2012). This study reported the presence of only α-T and δ-T3, but it lacked quantitative data regarding their concentrations. Furthermore, the specific plant parts of *H. perforatum* that were analyzed in this investigation were not disclosed [15]. Five years later, tocopherols, specifically α-T, γ-T, and δ-T, were detected in the upper portions of two-thirds of the examined St. John’s wort plants. Notably, δ-T was found to be predominant, and the total tocopherol content was reported to be less than 1 mg per 100 g dry weight (dw), which is considered atypical for leaves. It is important to note that this study did not account for the use of tocotrienol standards during the analysis [16]. In 2025, the presence and quantification of all four tocopherol and tocotrienol homologues (α, β, γ, and δ) were confirmed in the aerial parts of St. John’s wort, specifically within the upper 5 to 10 cm of the plant, which includes inflorescences along with some stems and leaves. Notably, tocotrienols were found to dominate over tocopherols in these inflorescences, with δ-T3 being particularly prevalent. This finding is significant, as it identifies *H. perforatum* as the first dicot species in the temperate climate zone to exhibit a predominance of δ-T3 among its tocochromanol constituents, a characteristic not observed in other plants within this region [14]. Tocochromanols occur in all photosynthetic organisms, however, α-T is the main form in photosynthetic tissues [18]. Therefore, due to the significant amounts of tocotrienols found in St. John’s wort inflorescences [14], it is worth looking closer at this plant and its lipophilic secondary metabolites. This is further compounded by the limited number (four) of reports on the tocopherol and tocotrienol contents in *H. perforatum* and certain limitations inherent in these studies, such as investigations focused solely on the leaves [16,17], plants’ analyzed anatomical parts not stated [15], or those examined as a single sample (flowers and flower buds with some stems and leaves) [14]. In those reports, the agronomical factors, such as year of harvest and location, were not investigated. Providing accurate information on the accumulation of tocotrienols in various aerial parts of St. John’s wort and their variability across different years of harvest could alter the model and potential applications of this medicinal plant. This could contribute not only to more effective utilization of *H. perforatum* but also to a more targeted and optimized approach. For example, a recent discovery demonstrated that hydroethanolic extraction is effective in recovering tocotrienols from St. John’s wort inflorescences [14].

The literature provides data describing the analgesic, anticancer, antibacterial, anti-aging, antihyperlipidemic, anticholesterolemic, antidiabetic, anti-inflammatory, antioxidant, antipyretic, and antithrombotic properties of tocotrienols [19,20,21]. In recent years, reports have highlighted the potent anticancer properties of tocotrienols against various types of cancer, including liver cancer, leukemia, breast cancer, prostate cancer, ovarian cancer, and lung cancer [21,22]. Studies by Neo et al. [23] demonstrated that tocotrienols may act as potential nootropic agents in enhancing memory. They improved synaptic plasticity by activating the BDNF/TrkB pathway. δ-T3 was identified as the homologue with the highest anticancer activity. Research by Sun et al. [24] revealed that δ-T3, as an immune enhancer, can disrupt PD-L1 glycosylation, thereby inhibiting tumor growth and improving the effectiveness of immunotherapy. Tocotrienols (γ-T3 and δ-T3) exhibit strong radioprotective effects by inducing the production of granulocyte colony-stimulating factor (G-CSF) in vivo. Metabolic studies by Zuo et al. [25] found that δ-T3 succinate had approximately seven times higher bioavailability compared to δ-T3 alone. It demonstrated superior efficacy in mitigating radiation-induced pancytopenia, enhancing the regeneration of hematopoietic stem and progenitor cells in the bone marrow, and promoting extramedullary hematopoiesis in the spleen of sublethally irradiated mice. Phytochemical studies of *Hypericum* plants have revealed the presence of polycyclic polyprenylated acylphloroglucinols (PPAPs), characterized by highly oxygenated acylphloroglucinol cores adorned with isoprenyl or geranyl groups. PPAPs exhibit a wide range of biological activities, including antidepressant, neuroprotective, memory-enhancing, cytotoxic, anti-inflammatory, antimicrobial, and antioxidant effects. Research by Wang et al. [26] also highlighted moderate cytotoxic activity against pancreatic cancer cell lines. Tocotrienols, which feature isoprenyl side chains, share structural similarities with the isoprenyl groups in PPAPs. Moreover, tocotrienols can integrate into cells more effectively than tocopherols due to their unsaturated side chains, which facilitate better penetration through cell membranes compared to the saturated chains of tocopherols [27].

There is very little information in the literature regarding the content of tocochromanols in *Hypericum*. To understand possible variations of tocotrienols concentrations in wild populations, St. John’s wort was collected in different locations over three years and separated into different aerial parts. Knowledge demonstrated in this study can help better understand the distribution and accumulation of tocotrienols in *H. perforatum*, the positive health potential of St. John’s wort, and the more efficient use of this plant in the food, pharmaceutical, and medical industries.

## 2. Results and Discussion

### 2.1. Tocochromanol Profile in Different Anatomical Parts of St. John’s Wort

At the beginning of the study, a screening of the tocopherol and tocotrienol profiles was conducted in the following parts of *H. perforatum*: roots, stems, leaves, flower buds, flowers, and seeds. The contents of tocochromanols in the stems were low, oscillating between 4 and 10 mg/100 g of tocochromanols, and including the content of tocotrienols at 2–5 mg/100 g. In the roots, only trace amounts of tocopherols were detected, while the seeds of *H. perforatum* were rich in γ-T (data not shown). Due to the low content of tocotrienols in stems and their absence in roots and seeds, they were excluded from the remaining (detailed) studies. The compositions of individual homologues of tocopherols and tocotrienols in leaves, flower buds, and flowers of wild *H. perforatum* are shown in Figure 1. All four homologues of tocopherols and tocotrienols were identified in leaves, flower buds, and flowers of St. John’s wort. The data obtained with the fluorescence detector were confirmed by the mass spectrometry (MS) study following the previous methodology [28]. Different anatomical parts varied in the composition of the examined compounds. The leaves were characterized by the highest composition of α-T (77%) and δ-T3 (18%), while the contents of other tocopherol and tocotrienol homologues did not exceed 5%. In flowers, the percentages of individual tocopherols and tocotrienols changed. The dominant homologue was δ-T3, which accounted for 42%, while the composition of α-T decreased to 38%. The compositions of other tocopherol and tocotrienol homologues also increased (totaling 20%). β-T, γ-T, and α-T3 were present in amounts greater than 5%. Analysis of tocopherol content in flower buds showed that they were characterized by the highest composition of tocotrienols, which accounted for 59% of the total tocochromanol content. Among tocopherols, α-T had the highest composition (30%), while among tocotrienols, α-T3 (22%) and δ-T3 (33%) were most abundant (Figure 1).

Tocopherols are present in leaves, but also in other plant parts, where they are synthesized and stored in plastids. The high composition of α-T is characteristic of green plant parts, as it better protects the photosynthetic apparatus from reactive oxygen species [18]. In turn, γ-T is the predominant tocochromanol found in the seeds of various plant species and their seed oils [29]. As reported by Horvath et al. [30], an analysis of tocopherol and tocotrienol content in over 80 different plant species led to the identification of only 22 species containing tocotrienols. Tocotrienols appear to be restricted to a small group of unrelated plants [30]. The lack of a clear taxonomic pattern among dicot species [30] may suggest that the ability to synthesize tocotrienols evolved independently in many plant families. However, certain chemotaxonomic relationships regarding the presence of tocotrienols in seed oils from various species belonging to the same family of dicotyledons have been observed [29]. A recent study on the tocopherol and tocotrienol profiles in the leaves of eleven *Hypericum* species reported significant quantities of tocotrienols within the *Hypericum* genus and suggested a potential connection to plant phylogeny [17].

### 2.2. Variability of Tocopherols and Tocotrienols Contents in St. John’s Wort: Impact of Plant Part and Harvest Year

Analyzing the total content of tocopherols and tocotrienols in the anatomical parts of *H. perforatum* collected in 2022–2024, it was found that the average tocopherol content in the leaves was 63.8 mg/100 g dw (41.6–96.7 mg/100 g dw), while the average tocotrienol content was 15.5 mg/100 g dw (9.9–23.6 mg/100 g dw). The calculated Ts/T3s ratio averaged 4.3 (1.8–8.0), clearly indicating the dominance of tocopherols in the studied leaves regardless of the year of collection. In flower buds, the opposite relationship was observed, with a higher content of tocotrienols, as evidenced by the Ts/T3s ratio of 0.7 (0.5–0.9). The total tocopherol content in flower buds was 42.9 mg/100 g dw (30.6–53.6 mg/100 g), while the tocotrienol content was found to be 20 mg/100 g dw higher than tocopherols. The average tocotrienol content in flower buds was 62.9 mg/100 g (45.1–81.6 mg/100 g dw). In the flowers, it was found that the content of tocopherols and tocotrienols was nearly equal—the Ts/T3s ratio averaged 1.1 (0.7–1.4). The total average tocopherol content was 38.1 mg/100 g dw (26.5–49.6 mg/100 g dw), while the average tocotrienol content in flowers was 35.5 mg/100 g dw (28.4–48.1 mg/100 g dw; Figure 2 and Figure 3 and Appendix A).

Appendix A) presents the results of the multifactorial analysis of variance for the studied tocopherols and tocotrienols, where the two analyzed factors were the year of collection and the anatomical part. The two-way analysis of variance revealed that for all tocopherol homologues, the year of collection and the anatomical part of the plant had a significant impact on their levels. For the α-T, interactions between the year of collection and the anatomical part were also observed. No such interactions were found for the other tocopherol homologues. In the analysis of tocotrienols, only the anatomical part of the plant had an effect. Interestingly, no effect of the year of collection on tocotrienol content was observed, regardless of the homologue. This confirmed that the content of this group of tocochromanols was only slightly influenced by environmental factors. For α-T3 and δ-T3, interactions between the year of collection and the anatomical part of the plant were observed (Appendix A). The anatomical part of St. John’s wort was characterized by statistically significant differences in the content of individual tocopherol and tocotrienol homologues (Appendix A). Table 1 presents the average content of tocopherol and tocotrienol homologues, broken down by anatomical parts. One-way analysis of variance revealed differences for all analyzed compounds. For α-T, the lowest content of these homologues was found in the flowers, followed by the flower buds, with the highest content in the leaves (27.9, 31.8, and 61.3 mg/100 g dw, respectively). For the other tocopherol homologues, statistically the least was found in the leaves, where their content did not exceed 2 mg/100 g dw. For flower buds and flowers, the tocopherol content was similar, with a slight predominance in the flower buds. In the case of tocotrienols, the leaves contained the statistically significantly (*p* < 0.05) lowest values, while the flower buds showed the highest levels. The content of α-T3 in the leaves and flowers was five and thirty-four times lower, respectively, than that recorded in the flower buds (23.7 mg/100 g dw). β-T3 was recorded in trace amounts (≤1 mg/100 g dw) in all plant parts, while γ-T3 was the most abundant in the flower buds (3.4 mg/100 g dw). The content of δ-T3 in the leaves was statistically significantly (*p* < 0.05) the lowest, while in the flower buds, it was the highest (14.5 and 34.8 mg/100 g dw, respectively). Flowers and flower buds were characterized by similar concentrations of δ-T3, but statistically significant (*p* < 0.05; Table 1).

Multifactorial analysis of variance revealed that the year of collection was significant only in the case of tocopherols. Table 2 presents the results concerning the differences in their content in the different years. α-T showed the highest content in 2022 and 2024 (39.9 and 43.4 mg/100 g dw, respectively). The lowest content of this homologue was recorded in 2023 (37.7 mg/100 g dw). The other tocopherol homologues also varied between years of collection, but their amounts did not exceed 5 mg/100 g dw (Table 2).

Table 3 presents the interactions between the year of collection and the anatomical parts of St. John’s wort, which were observed for the contents of α-T, α-T3, and δ-T3. For α-T, it was shown that its content was statistically significantly the highest in the leaves, where significant differences were found between the years of collection. In 2022, the leaves contained 59.9 mg/100 g dw of α-T. In the following year, a slight decrease occurred to 54.1 mg/100 g dw. In 2024, a significant increase was observed, with α-T content rising to 72.7 mg/100 g dw. For α-T3, the highest content of this tocotrienol was found in the flower buds (21.5–25.4 mg/100 g), with the flower buds from 2022 showing the highest content. For δ-T3, the leaves exhibited the lowest content (13.3 to 16.1 mg/100 g dw), and the year of collection did not affect the content. Greater variation was observed in the flower buds and flowers, where the contents ranged from 30 to 36.2 mg/100 g dw, with differences observed between the years of collection as well as between flower buds and flowers (Table 3 and Figure 3).

The biosynthetic pathways of tocopherols and tocotrienols have been extensively described in the literature; however, the highly diverse plant world continues to surprise, revealing its complexity in determining the production of tocotrienols/tocopherols. Tocotrienols are found, sometimes alongside tocopherols, in specific tissues (e.g., seeds and latex) and certain plant species (e.g., palm oils) [31,32,33]. In our study, tocotrienols dominated in the flower buds and flowers of *H. perforatum*. As Muñoz and Munné-Bosch [31] suggested, the uneven distribution of tocotrienols in the plant kingdom and the fact that they do not naturally replace tocopherols indicate that tocopherols are essential compounds widely distributed throughout the plant kingdom, whereas tocotrienols act as secondary metabolites.

### 2.3. The Potential Applications, Challenges, and Future Directions for Tocotrienol-Rich Extracts of H. perforatum

Unlike tocopherols, tocotrienols exhibit strong anticancer effects by inhibiting angiogenesis [34]. The studied anatomical parts of St. John’s wort, particularly flower buds and flowers, could serve as raw materials for the production of preparations rich in tocotrienols. This is especially relevant since their content in this raw material is stable and less affected by environmental factors compared to tocopherols. Additionally, it has been reported that tocotrienols and tocopherols can be effectively extracted using hydroethanolic solutions (pharmaceutical-grade solvents) from *H. perforatum* inflorescences [14]. Several reports indicate the significant therapeutic potential of tocotrienols, particularly in the treatment of neurodegenerative diseases, cancer, and chronic inflammatory conditions, such as diabetic nephropathy. However, the study performed by Shibata et al. [35] highlighted the challenges associated with the interaction between tocopherols and tocotrienols, which may affect the efficacy of preparations. The simultaneous administration of α-T reduced the uptake of δ-T3 in human colorectal adenocarcinoma DLD-1 cells in a dose-dependent manner. The δ-T3-induced cell cycle arrest and the expression of pro-apoptotic genes/proteins (e.g., p21, p27, and caspases) were negated by α-T [35]. The co-presence of α-T may reduce the uptake of δ-T3 in cancer cells and attenuate its pro-apoptotic effects, suggesting that tocotrienol-based preparations may need to be purified of tocopherols. For the reasons outlined above, it appears advisable to exclude the leaves from the preparation of tocotrienol-rich extracts due to the dominant presence of α-tocopherol (77%). In contrast, the situation is more favorable for flower buds and flowers, which contain 30% and 38% of α-T, respectively. However, even in these cases, separating tocotrienols from tocopherols, mainly α-T, may pose significant challenges, particularly from an economic standpoint.

Further research should also focus on the precise mechanisms of action of tocotrienols, such as δ-T3, in different disease models, which could contribute to the development of more targeted therapies. For example, findings by Matsura et al. [36] suggested that δ-T3 exerts cytoprotective effects in Parkinson’s disease models through the ERβ/PI3K/Akt pathway, which requires further validation in clinical trials. In addition, the antioxidant and anti-inflammatory properties of tocotrienols suggest their potential in the treatment of diabetic complications, such as nephropathy, as demonstrated in studies in diabetic rats [37].

Another critical aspect for future research and industrial applications is the possibility of using tocotrienols as agents to sensitize cancer cells to chemotherapy, potentially enhancing the efficacy of standard anticancer therapies. As noted by Husain and Malafa [38], tocotrienols modulate key signaling molecules, such as NF-κB and STAT-3, opening up new possibilities in oncological treatment.

The data obtained in this study may also serve as a basis for the development of cultivation standards for St. John’s wort, which would increase the efficiency of the production of biologically active compounds. For example, optimizing the timing of harvesting could increase the levels of desired compounds in selected parts of the plant, such as flowers or flower buds, which are characterized by the highest levels of tocotrienols. In addition, the selective use of certain parts of the plant allows better utilization of the plant material, reducing waste and improving the economic efficiency of the production process. In addition, St. John’s wort can serve as a multifunctional plant, both as a source of biologically active compounds and as a nectar-producing plant that supports pollinator populations and biodiversity. This is particularly important in the context of environmental protection and climate change, as it can contribute to the implementation of more sustainable agricultural practices. On the other hand, St. John’s wort cultivation can be a valuable source of income for farmers, especially in less developed regions. The high added value of pharmaceutical and cosmetic products derived from St. John’s Wort increases the attractiveness of this plant as an industrial crop.

## 3. Materials and Methods

### 3.1. Reagents

Ethanol, methanol, ethyl acetate, *n*-hexane (HPLC grade), pyrogallol, sodium chloride, and potassium hydroxide (reagent grade) were purchased from Sigma-Aldrich (Steinheim, Germany). Ethanol (96.2%) for leaf samples’ saponification was received from SIA Kalsnavas Elevators (Jaunkalsnava, Latvia). Standards of tocopherol homologues (α, β, γ, and δ; >98%, HPLC) were obtained from Extrasynthese (Genay, France), while tocotrienol homologues (α, β, γ, and δ; >98%, HPLC) were from Cayman Chemical (Ann Arbor, MI, USA).

### 3.2. Plant Material

Nine various populations of wild *H. perforatum* were harvested each year during 2022–2024 at three different places, Gawrony (52°02′30.7″ N 17°02′13.3″ E), Środa (52°13′39.9″ N, 17°18′09.7″ E), and Śrem (52°05′19.3″ N 17°01′49.5″ E), in Poland, located within a radius of each other of 5 to 25 km. In each location, three populations of wild St. John’s wort, located at least 0.5 km from each other, were harvested. Wild *H. perforatum* was identified according to a taxonomic guide (https://powo.science.kew.org, accessed on 10 June 2022) by Dr. Paweł Górnaś (PhD), who was previously trained by several botanists prior to the collection of the plants. The aerial parts of *H. perforatum* are presented in Appendix A). Wild *H. perforatum* was collected at full bloom when plants contained both flower buds and flowers. Each of the 9 biological replications of wild St. John’s wort were collected randomly (15–25 plants) by cutting 5–10 cm from the soil and separated into 4 aerial parts (stems, leaves, flower buds, and flowers). Separate parts were pre-dried in opened paper boxes at room temperatures of 24 ± 3 °C for 7 ± 1 days, during car transport and until the final destination point (laboratory). To remove the residues of water, plant material was finally freeze-dried using a FreeZone freeze-dry system (Labconco, Kansas City, MO, USA) at a temperature of −51 ± 1 °C under vacuum of below 0.01 mbar for 48 h. Freeze-dried aerial parts (2–10 g each) were powdered using an MM 400 mixer mill (Retsch, Haan, Germany) and directly used for sample preparation according to the method below. The remnants of powdered samples were transferred into polypropylene bags and stored at −18 °C. The dry mass was measured gravimetrically.

### 3.3. Saponification and n-Hexane:Ethyl Acetate Extraction Protocol

The saponification protocol was performed as described earlier [39]. An amount of 0.1 g of the powdered leaf sample was placed in a 15 mL glass tube with a screw cap. Then, 0.05 g of pyrogallol was added to prevent oxidation of tocopherols and tocotrienols. The mixture was sequentially supplemented with 2.5 mL of 96.2% ethanol and mixed. The process of saponification was incited by adding 0.25 mL of 60% (*w*/*v*) aqueous potassium hydroxide. The glass tube was immediately closed with a screw cap and mixed for 10–15 s using the vortex REAX top (Heidolph, Schwabach, Germany), with vibration frequency rates up to 2500 rpm, and sequentially subjected to incubation in a water bath at 80 °C. After 10 min of incubation, the sample was mixed again for 10–15 s using the vortex REAX top at 2500 rpm. After 25 min of incubation to stop/slow down the process of saponification, the sample was cooled immediately in an ice-water bath for 10 min. The process of tocopherol and tocotrienol homologues’ extraction started by adding 2.5 mL of 1% (*w*/*v*) sodium chloride to the glass tube with the sample to lower the surface tension between the two non-miscible solvents (hydro-ethanol and *n*-hexane:ethyl acetate), and it was mixed for 5 s using the vortex REAX top at 2500 rpm. Then, 2.5 mL of *n*-hexane:ethyl acetate (9:1; *v*/*v*) was added to extract the tocopherol and tocotrienol homologues, and mixed for 15 s using the vortex REAX top at 2500 rpm. After mixing with the organic solvent mixture (*n*-hexane and ethyl acetate), the sample was centrifuged for 5 min (1000× *g*, at 4 °C). The organic layer, containing *n*-hexane and ethyl acetate, was moved to a 100 mL round-bottom flask. The extraction residues were re-extracted in a fresh portion of 2.5 mL of *n*-hexane:ethyl acetate (9:1; *v*/*v*), as described above. Re-extraction was performed two times. The organic layers from the initial extraction and two re-extractions were collected and combined in the same 100 mL round-bottom flask and evaporated in a vacuum rotary evaporator, Laborota 4000 (Heidolph, Schwabach, Germany), at 40 °C until fully dry. The obtained thin-film layer on the bottom of the flask was dissolved in 1 mL of ethanol (HPLC grade) and transferred to a 2 mL analytical glass vial.

### 3.4. Tocopherol and Tocotrienol Determination by RP-HPLC-FLD

The tocochromanol analysis was performed using reverse-phase high-performance liquid chromatography with a fluorescent light detector (RP-HPLC-FLD) via the HPLC Shimadzu Nexera 40 Series system (Kyoto, Japan), consisting of a pump (LC-40D pump), a degasser (DGU-405), a system controller (CBM-40), an autoinjector (SIL-40C), a column oven (CTO-40C), and a fluorescence detector (RF-20Axs). The chromatographic separation of tocopherol and tocotrienol homologues was carried out using the Epic PFP-LB column with a pentafluorophenyl phase (PerkinElmer, Waltham, MA, USA). The column parameters included fully porous particle morphology, a particle size of 3 µm, a column length of 150 mm, and a column internal diameter of 4.6 mm. A guard column was used, measuring 4 mm in length and 3 mm in internal diameter (Phenomenex, Torrance, CA, USA). The chromatography analysis was performed under isocratic conditions with the following specifications: the mobile phase consisted of methanol and water (91:9; *v*/*v*), the flow rate was 1.0 mL/min, the column oven temperature was 45 ± 1 °C, and the room temperature was 21 ± 1 °C. The total chromatography runtime was 13 min. Identification and quantification were conducted using a fluorescence detector with an excitation wavelength of 295 nm and an emission wavelength of 330 nm. Quantification was performed based on calibration curves obtained from tocopherol and tocotrienol standards.

### 3.5. LC-MS Presence Confirmation of Tocochromanols in St. John’s Wort Aerial Parts

The results of the presence of tocopherols and tocotrienols in leaves, flowers, and flower buds of St. John’s wort obtained by RP-HPLC-FLD analysis were confirmed by LC-MS analysis according to the previously developed method [28]. Briefly, chromatographic separation was performed on a Kinetex PFP column (1.7 µm, 100 × 3 mm; Phenomenex, Torrance, CA, USA) using a binary mobile phase system consisting of water (A) and methanol (B) via an UltiMate 3000 HPLC system (Dionex, Sunnyvale, CA, USA). The gradient elution program was as follows: hold at 20% B for 1 min; 1–9.5 min, linear increase from 20% to 95% B; 9.5–25 min, hold at 95% B; 25–25.1 min return to 20% B, 25.1–28 min, hold at 20% B. The system was run at a flow rate of 0.3 mL/min. The mass spectra were obtained using a Q-Exactive Orbitrap MS system (Thermo Scientific, Dreieich, Germany) equipped with an IonMax atmospheric pressure chemical ionization (APCI) source in positive and negative ionization mode. Quantification of tocopherols and tocotrienols was carried in full-scan mode at a resolution of 70,000 FWHM (at *m*/*z* 200) from 100 to 1000 *m*/*z*. For these measurements, negative APCI mode was employed. Compounds were measured by comparing peak areas of corresponding [M^−^ H]^−^ ions with a ±5 ppm mass accuracy tolerance and a ±0.1 min retention time tolerance. The limit of quantification (LoQ) was determined by analyzing a standard solution at low concentrations ranging from 0.01 to 0.25 ng/μL. The lowest concentration that resulted in chromatographic peaks with a signal-to-noise ratio of ≥10 was chosen as the LoQ. Mass accuracy of the Orbitrap HRMS system (<5 ppm) was maintained via external calibration using Pierce LTQ ESI Positive Ion Calibration Solution and Pierce ESI Negative Ion Calibration Solution (Thermo Scientific, Dreieich, Germany). Data acquisition and processing were conducted using Thermo Scientific Xcalibur (v. 4.1).

### 3.6. Statistical Analysis

Data visualization was carried out using Excel (Version 2302) from Microsoft 365 Apps for enterprise (Redmond, WA, USA) and R (R version 4.3.3 (2024-02-29 ucrt)) (R Foundation for Statistical Computing, Vienna, Austria), an open-source software. To evaluate the statistical influence of the plant parts and harvest year on the tocochromanol profile, the analysis was performed using Statistica 10.0 (StatSoft. Inc., Tulsa, OK, USA). A *p*-value of ≤0.05 was considered indicative of significant differences between mean values, as determined by a multi-factorial analysis of variance (ANOVA). For significant findings, Tukey’s post hoc tests were used to identify homogeneous groups. The graphical tool heatmap was used to visualize the data, allowing the presentation of differences in the concentrations of bioactive compounds depending on plant parts and harvest years. The heatmap was developed based on the average concentration values for each bioactive compound, taking into account the averaged data for each year. The analyses were conducted using statistical and visualization libraries in Python (including Seaborn and Matplotlib).

## 4. Conclusions

The content of tocochromanol in different anatomical parts of St. John’s wort depended on both the year of collection and the specific plant part. Tocopherols predominated in the leaves, while tocotrienols were present in higher concentrations in the flowers and flower buds, suggesting their distinct role in the plant’s metabolism. Studies indicated that the year of collection had a significant impact on the content of tocopherols, while tocotrienol content showed less sensitivity to environmental variability, which may suggest their higher stability. *H. perforatum* can be a valuable source of biologically active compounds, particularly tocotrienols, which are characterized by higher stability and less environmental influence on their content. This could be an important factor for future field production of St. John’s wort, especially with the aim of its use in the cosmetic or pharmaceutical industries.

The results of our study indicated that differences in the tocochromanol content between various parts of the plant and their stability under variable environmental conditions could form the basis for developing more sustainable cultivation practices for *H. perforatum*. Precisely determining the optimal harvest time and utilizing selected plant parts allows for the optimization of biologically active compound production while simultaneously minimizing waste and environmental impacts. *H. perforatum* could thus serve as a model example of a medicinal plant whose cultivation contributes to biodiversity conservation and the implementation of sustainable agricultural practices. The findings may also support the development of cultivation systems for medicinal plants aimed at improving the predictability of both the quality and quantity of obtained raw materials, which is crucial for the pharmaceutical and cosmetic industries.

Future research should investigate the genetic basis of tocotrienol accumulation in *H. perforatum* to identify genetic and/or biotechnological enhancement opportunities. The next step would be to develop sustainable and economically viable extraction and purification techniques aimed at obtaining high-quality products rich in health-promoting bioactive compounds. Additionally, exploring the synergistic effects of tocotrienols with other bioactive phytochemicals in St. John’s wort, e.g., hypericin and hyperforin, could lead to combined therapeutic applications. Finally, clinical studies are necessary to validate the pharmacological efficacy of the extracted bioactive molecules, particularly δ-T3.

## Figures and Tables

**Figure 1 molecules-30-00901-f001:**
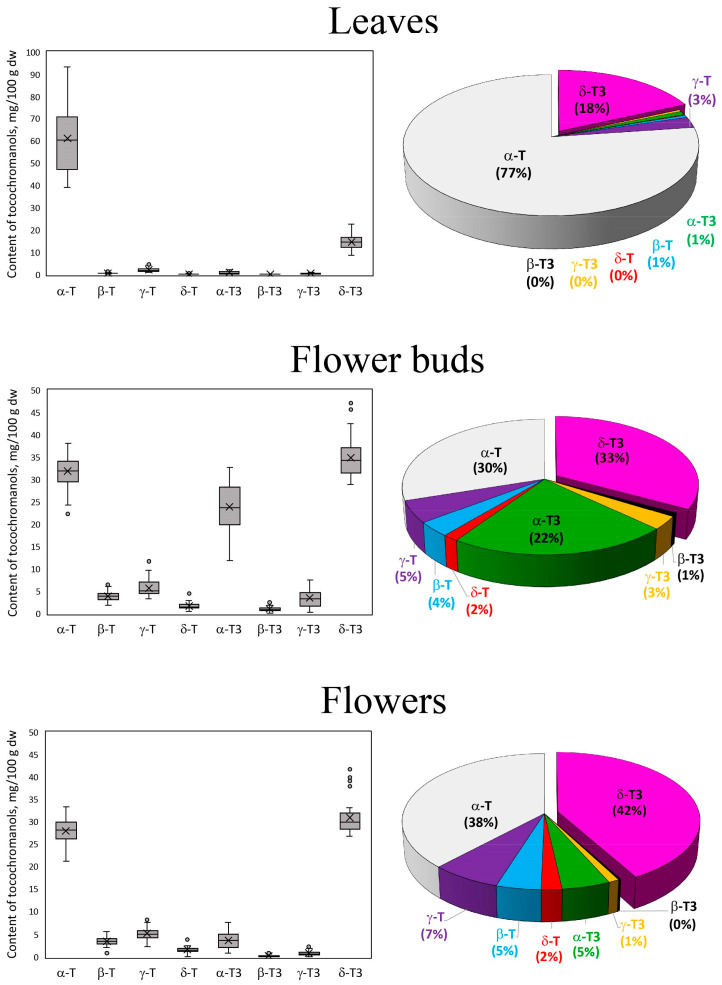
The box plots of the content (mg/100 g dw) and average content proportion (%) of tocotrienol (T3) and tocopherol (T) homologues (α, β, γ, and δ) in leaves, flower buds, and flowers of wild *H. perforatum* populations during the years 2022–2024.

**Figure 2 molecules-30-00901-f002:**
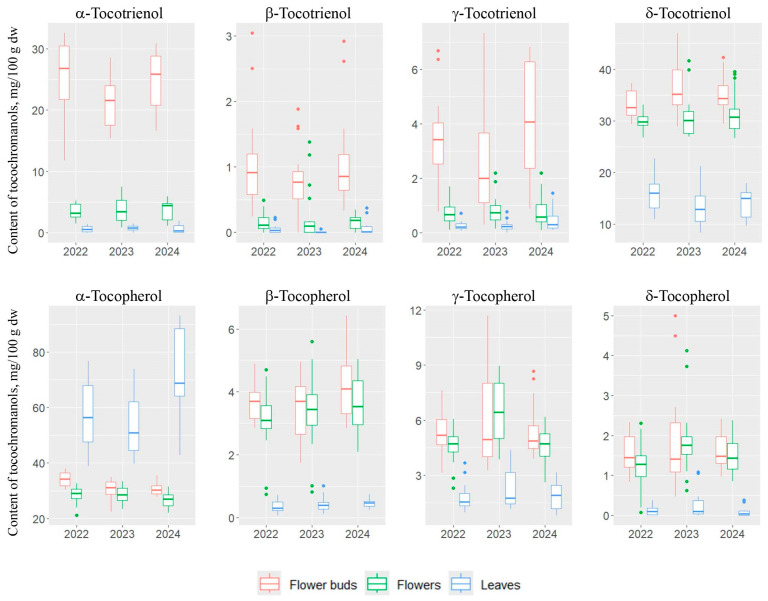
The box plots of the content (mg/100 g dw) variability of tocotrienol (T3) and tocopherol (T) homologues (α, β, γ, and δ) in leaves, flower buds, and flowers in each year of study (2022–2024).

**Figure 3 molecules-30-00901-f003:**
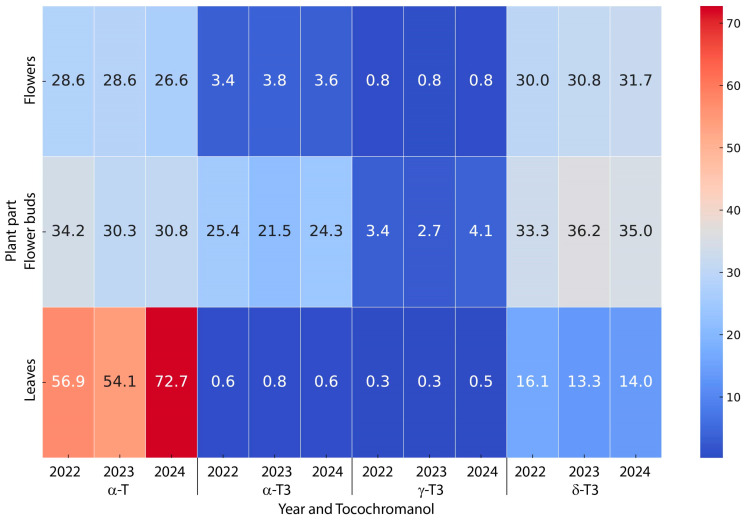
Heatmap illustrating the average concentrations (mg/100 g dw) of the main tocochromanols (α-T, α-T3, γ-T3, and δ-T3) in leaves, flower buds, and flowers across harvest years (2022–2024).

**Table 1 molecules-30-00901-t001:** Analysis of variance for tocochromanol contents (mg/100 g dw) in three aerial parts (leaves, flower buds, and flowers) of wild *H. perforatum* harvested during 2022–2024.

Tocochromanol	Plant Part		
	Leaves	Flower Buds	Flowers
α-T	61.3 ± 16.0 ^c^	31.8 ± 3.4 ^b^	27.9 ± 3.0 ^a^
β-T	0.4 ± 0.2 ^a^	3.8 ± 1.0 ^c^	3.4 ± 1.0 ^b^
γ-T	2.0 ± 0.9 ^a^	5.6 ± 1.8 ^b^	5.2 ± 1.5 ^b^
δ-T	0.1 ± 0.2 ^a^	1.7 ± 0.8 ^b^	1.5 ± 0.7 ^b^
α-T3	0.7 ± 0.5 ^a^	23.7 ± 5.3 ^c^	3.6 ± 1.6 ^b^
β-T3	0.0 ± 0.1 ^a^	1.0 ± 0.7 ^b^	0.2 ± 0.3 ^a^
γ-T3	0.3 ± 0.3 ^a^	3.4 ± 2.0 ^b^	0.8 ± 0.5 ^a^
δ-T3	14.5 ± 3.6 ^a^	34.8 ± 4.0 ^c^	30.8 ± 3.5 ^b^

Values are expressed as the mean ± standard deviation (*n* = 9). Different letters in the same line indicate a statistically significant difference at *p* ≤ 0.05. T, tocopherol; T3, tocotrienol.

**Table 2 molecules-30-00901-t002:** Year ANOVA for tocochromanol contents (mg/100 g dw) in three aerial parts (leaves, flower buds, and flowers) of wild *H. perforatum* harvested during 2022–2024.

Tocochromanol	Year		
	2022	2023	2024
α-T	39.9 ± 14.5 ^ab^	37.7 ± 13.6 ^a^	43.4 ± 23.2 ^b^
β-T	2.4 ± 1.6 ^a^	2.4 ± 1.7 ^ab^	2.8 ± 1.9 ^b^
γ-T	3.9 ± 1.9 ^a^	5.0 ± 2.7 ^b^	4.0 ± 1.9 ^a^
δ-T	1.0 ± 0.8 ^a^	1.3 ± 1.2 ^b^	1.1 ± 0.8 ^ab^

Values are expressed as the mean ± standard deviation (*n* = 12). Different letters in the same line indicate a statistically significant difference at *p* ≤ 0.05. T, tocopherol.

**Table 3 molecules-30-00901-t003:** Interaction effects of plant parts and year on tocochromanol contents (mg/100 g dw) in three aerial parts (leaves, flower buds, and flowers) of wild *H. perforatum* harvested during 2022–2024.

Plant Part	Year	Tocochromanol		
		α-T	α-T3	δ-T3
Leaves	2022	59.9 ± 12.7 ^b^	0.6 ± 0.5 ^a^	16.1 ± 3.7 ^a^
	2023	54.1 ± 11.3 ^b^	0.8 ± 0.4 ^a^	13.3 ± 3.7 ^a^
	2024	72.7 ± 17.1 ^c^	0.6 ± 0.6 ^a^	14.0 ± 2.8 ^a^
Flower buds	2022	34.2 ± 2.4 ^a^	25.4 ± 5.9 ^c^	33.3 ± 2.7 ^bcd^
	2023	30.3 ± 3.7 ^a^	21.5 ± 4.2 ^b^	36.2 ± 5.2 ^d^
	2024	30.8 ± 2.5 ^a^	24.3 ± 5.0 ^bc^	35.0 ± 3.4 ^cd^
Flowers	2022	28.6 ± 3.0 ^a^	3.4 ± 1.3 ^a^	30.0 ± 1.6 ^b^
	2023	28.6 ± 2.3 ^a^	3.8 ± 2.0 ^a^	30.8 ± 4.1 ^b^
	2024	26.6 ± 2.8 ^a^	3.6 ± 1.6 ^a^	31.7 ± 4.2 ^bc^

Values are expressed as the mean ± standard deviation (*n* = 3). Different letters in the same column indicate a statistically significant difference at *p* ≤ 0.05. T, tocopherol; T3, tocotrienol.

## Data Availability

The data used to support the findings of this study are available in Appendix A and from the corresponding author upon request.

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
