# Peer review of "Rare Prenyllipids in Wild St. John’s Wort During Three Harvest Seasons"

_molecules, 2025, doi:10.3390/molecules30040901_

Round 1

Reviewer 1 Report

Comments and Suggestions for Authors

The study focuses on wild populations from a geographically limited area in Poland. Expanding the scope to include populations from different climatic regions would enhance the generalizability of the findings. Comparing populations from diverse climates could provide valuable insights into the environmental effects on tocotrienol and tocopherol concentrations.

While the three-year study period offers valuable insights, it may not fully account for long-term trends or anomalies. Extending the study to include additional harvest years would allow for a more comprehensive exploration of long-term environmental effects.

The study exclusively analyzes aerial parts of the plant, potentially overlooking bioactive compounds present in roots or seeds. Including other plant parts in the analysis would provide a more holistic understanding of compound distribution.

Tocopherols reduce the efficacy of tocotrienols, necessitating purification, and increasing production costs. To address this, the study could explore cost-effective methods for tocotrienol extraction or strategies to mitigate tocopherol interference.

The claim that tocotrienols are less sensitive to environmental factors is intriguing but requires further verification. A mechanistic explanation or hypothesis for their observed stability would strengthen this assertion.

The number and diversity of sampling sites should be increased to include regions with varying soil types, altitudes, and climatic conditions, ensuring broader applicability of the results.

The use of RP-HPLC-FLD and LC-MS methodologies is robust; however, incorporating additional techniques, such as NMR spectroscopy, could further validate the identification and structural confirmation of the compounds.

While the tables and box plots are informative, additional visual tools like heat maps could better illustrate variations across plant parts and collection years.

The introduction provides a strong background but could better emphasize the study’s novelty, mainly how it builds upon previous research.

The detailed results section would benefit from a more apparent distinction between findings and interpretation. Subheadings could organize the results by anatomical part and year.

The discussion occasionally overlaps with the results. To avoid redundancy, these sections should be more distinctly separated. Additionally, the debate could be expanded to highlight the practical applications of the findings, particularly in the pharmaceutical and cosmetic industries.

The conclusion effectively summarizes key findings but could emphasize broader implications, such as potential contributions to sustainable farming practices for medicinal plants.

Future research should explore the genetic basis of tocotrienol and tocopherol production in H. perforatum to identify genetic or biotechnological enhancement opportunities. Investigating the synergistic effects of tocotrienols with other bioactive compounds in St. John’s Wort could lead to combined therapeutic applications. Finally, clinical studies are needed to validate the pharmacological efficacy of the extracted compounds, particularly δ-T3.

Author Response

We sincerely thank you for all the comments, remarks, and suggestions that have contributed to enhancing the manuscript and its scientific quality. The manuscript and supplementary materials have been improved accordingly. Provided changes are marked in red font. For literature we used references manager software therefore changes are not highlighted.

Reviewer 1

Comments 1: The study focuses on wild populations from a geographically limited area in Poland. Expanding the scope to include populations from different climatic regions would enhance the generalizability of the findings. Comparing populations from diverse climates could provide valuable insights into the environmental effects on tocotrienol and tocopherol concentrations.

Response 1: Thank you for the comment. We agree that such an approach could provide a broader context for the analyses and allow generalisation of the results. However, our study was primarily focused on a detailed examination of wild populations from a specific region of Poland, which allowed us to perform precise analyses under relatively uniform environmental conditions. It should be emphasised that plant samples were collected over several years (2022-2024), which allowed us to capture potential seasonal variations. Further research, including populations from different climatic regions, is planned for the future.

Comments 2: While the three-year study period offers valuable insights, it may not fully account for long-term trends or anomalies. Extending the study to include additional harvest years would allow for a more comprehensive exploration of long-term environmental effects.

Response 2: Thank you for the comment. A longer observation period could enable a more comprehensive analysis of long-term trends and potential anomalies. Nevertheless, the three-year study period (2022–2024) is commonly used in similar environmental studies, particularly in the context of analyzing seasonality or the influence of environmental factors on organisms. This timeframe allows for capturing variability between seasons as well as the effects of short-term environmental changes, such as differences in temperature or precipitation between years. Additionally, we received funding to conduct research on the topic of Hypericum for only three years, and we must conclude the project by presenting the results within this timeframe.

Comments 3: The study exclusively analyzes aerial parts of the plant, potentially overlooking bioactive compounds present in roots or seeds. Including other plant parts in the analysis would provide a more holistic understanding of compound distribution.

Response 3: Thank you for the comment. We agree that the inclusion of other plant parts, such as roots or seeds, could provide a more comprehensive understanding of the distribution of bioactive compounds. However, our study focuses on the aerial parts of plants as these are the most commonly analysed in relation to bioactive compound content due to their widespread use in chemical, pharmacological and nutritional studies. The green parts of plants also represent the primary site of synthesis and accumulation of the compounds analysed, making them a key focus of research. The roots and seeds were analyzed; however, due to the fact that they did not contain tocotrienols and are not utilized in herbal medicine, we did not continue their analysis. Nevertheless, to enhance the manuscript with information regarding tocochromanols in the roots and seeds, we included the following statement in the manuscript: “In the roots, only trace amounts of tocopherols were detected, while the seeds of H. perforatum were rich in γ-T (data not shown). Due to the low content of tocotrienols in stems and their absence in roots and seeds, they were excluded from the remaining (detailed) studies.” (Page 3, bottom part).

Comments 4: Tocopherols reduce the efficacy of tocotrienols, necessitating purification, and increasing production costs. To address this, the study could explore cost-effective methods for tocotrienol extraction or strategies to mitigate tocopherol interference.

Response 4: Thank you for the comment. The obtained results provide important insights into the potential of the plants studied in our research as sources of tocotrienols. These data can serve as a solid foundation for future studies that could focus on the development of more cost-effective extraction methods and strategies to reduce tocopherol interference, which undoubtedly represents an interesting direction for further research. Economic evaluation of the profitability of extraction, purification, and application requires separate studies. Our study is intended to lay the foundation for future extraction and purification studies. However, a short statement about extraction, from our previous study was added: „Additionally, it has been reported that tocotrienols and tocopherols can be effectively extracted using hydroethanolic solutions (pharmaceutical-grade solvent) from H. perforatum inflorescences [14].” (Page 9, top part) and “The next step would be to develop sustainable and economically viable extraction and purification techniques aimed at obtaining high-quality products rich in health-promoting bioactive compounds.” (Page 12, bottom part).

Comments 5: The claim that tocotrienols are less sensitive to environmental factors is intriguing but requires further verification. A mechanistic explanation or hypothesis for their observed stability would strengthen this assertion.

Response 5: Thank you for the comment. The sentence “This confirms that the content of this group of tocopherols is only slightly influenced by environmental factors” has been removed from the manuscript. In our opinion, additional studies should be conducted to confirm the validity of this claim.

Comments 6: The number and diversity of sampling sites should be increased to include regions with varying soil types, altitudes, and climatic conditions, ensuring broader applicability of the results.

Response 6: Thank you for the comment. In the present study, we focused on a more in-depth analysis within a limited area, which allows for a better understanding of local mechanisms and detailed tocochromanol profile in aerial parts of H. perforatum used for medical purposes.

Comments 7: The use of RP-HPLC-FLD and LC-MS methodologies is robust; however, incorporating additional techniques, such as NMR spectroscopy, could further validate the identification and structural confirmation of the compounds.

Response 7: Thank you for the comment. We agree that NMR is a valuable tool in structural studies that could provide additional information. Nevertheless, in this study, the primary methods used were RP-HPLC-FLD and LC-MS, which are widely recognized and applied in the quantitative and qualitative analysis of tocopherols and tocotrienols. The combination of these two techniques offered both high sensitivity and selectivity, enabling the identification and precise determination of the concentrations of the analyzed compounds. We would like to point out that in both techniques tocopherol and tocotrienol standards were used on two different systems by two different persons. MS confirmation was performed not only identification but also separation using previously developed MS method. Additionally, the introduction part contains information that the presence of δ-T3, main tocotrienol, was already detected in St. John's Wort over a decade ago via MS. “The identification of tocopherol and tocotrienol homologues in various medicinal plants, including St. John's Wort, was first accomplished using electrospray ionization coupled with liquid chromatography-tandem mass spectrometry (ESI(+)-LC-MS/MS) over a decade ago (2012).” (Page 2, middle part).

Comments 8: While the tables and box plots are informative, additional visual tools like heat maps could better illustrate variations across plant parts and collection years.

Response 8: Thank you for the comment. In accordance with the reviewer’s comments, we have created heatmap for the obtained results to better illustrate them. We have prepared heatmaps only for the variables with the highest content in the analyzed samples (δ-T3, γ-T3, α-T3, α-T) for easy read. The data of all detected tocochromanols is presented in the Supplementary Materials. We have also supplemented the statistical description. (Page 6, bottom part, Page 12, middle part).

Comments 9: The introduction provides a strong background but could better emphasize the study’s novelty, mainly how it builds upon previous research.

Response 9: Thank you for the comment. The following paragraph was added: “This is further compounded by the limited number (four) of reports on the tocopherol and tocotrienol content in H. perforatum and certain limitations inherent in these studies such as investigation focused solely on the leaves [16, 17], plant's analyzed anatomical parts not stated [15], or examined as a single sample (flowers and flower buds with some stems and leaves) [14]. In any of those reports, the agronomical factors such as year of harvest and location were not investigated. Providing accurate information on the accumulation of tocotrienols in various aerial parts of St. John's wort and their variability across different years of harvest could alter the model and potential applications of this medicinal plant. This could contribute not only to more effective utilization of H. perforatum but also to a more targeted and optimized approach. For example, the recent discovery demonstrates that hydroethanolic extraction is effective in recovering tocotrienols from St. John's wort inflorescences [14].” (Page 2, bottom part, Page 3, top part).

Comments 10: The detailed results section would benefit from a more apparent distinction between findings and interpretation. Subheadings could organize the results by anatomical part and year. The discussion occasionally overlaps with the results. To avoid redundancy, these sections should be more distinctly separated. Additionally, the debate could be expanded to highlight the practical applications of the findings, particularly in the pharmaceutical and cosmetic industries.

Response 10: Thank you for the comment. We have a somewhat different perspective; however, we have made a number of changes to enhance the readability of the manuscript. Three subparagraphs were added and some information relocation was made. (Page 3, bottom part, Page 5, whole page, Page 7, middle part, Page 9, whole page, Page 10, top part).

Comments 11: The conclusion effectively summarizes key findings but could emphasize broader implications, such as potential contributions to sustainable farming practices for medicinal plants.

Response 11: Thank you for the comment. The conclusions part has been extended and supplemented: “The results of our study indicate that differences in the tocochromanols content between various parts of the plant and their stability under variable environmental conditions could form the basis for developing more sustainable cultivation practices for H. perforatum. Precisely determining the optimal harvest time and utilizing selected plant parts allows for the optimization of biologically active compound production while simultaneously minimizing waste and environmental impact. H. perforatum could thus serve as a model example of a medicinal plant whose cultivation contributes to biodiversity conservation and the implementation of sustainable agricultural practices. The findings may also support the development of cultivation systems for medicinal plants aimed at improving the predictability of both the quality and quantity of obtained raw materials, which is crucial for the pharmaceutical and cosmetic industries. Future research should investigate the genetic basis of tocotrienol accumulation in H. perforatum to identify genetic and/or biotechnological enhancement opportunities. The next step would be to develop sustainable and economically viable extraction and purification techniques aimed at obtaining high-quality products rich in health-promoting bioactive compounds. Additionally, exploring the synergistic effects of tocotrienols with other bioactive phytochemicals in St. John’s wort, e.g. hypericin and hyperforin, could lead to combined therapeutic applications. Finally, clinical studies are necessary to validate the pharmacological efficacy of the extracted bioactive molecules, particularly δ-T3.” (Page 12, bottom part, Page 13, top part).

Comments 12: Future research should explore the genetic basis of tocotrienol and tocopherol production in H. perforatum to identify genetic or biotechnological enhancement opportunities. Investigating the synergistic effects of tocotrienols with other bioactive compounds in St. John’s Wort could lead to combined therapeutic applications. Finally, clinical studies are needed to validate the pharmacological efficacy of the extracted compounds, particularly δ-T3.

Response 12: Thank you for the comment. We agree. This is an excellent proposal for a future multifaceted research project. In our conclusions, we have included suggestions for future studies. “Future research should investigate the genetic basis of tocotrienol accumulation in H. perforatum to identify genetic and/or biotechnological enhancement opportunities. The next step would be to develop sustainable and economically viable extraction and purification techniques aimed at obtaining high-quality products rich in health-promoting bioactive compounds. Additionally, exploring the synergistic effects of tocotrienols with other bioactive phytochemicals in St. John’s wort, e.g. hypericin and hyperforin, could lead to combined therapeutic applications. Finally, clinical studies are necessary to validate the pharmacological efficacy of the extracted bioactive molecules, particularly δ-T3.” (Page 12, bottom part, Page 13, top part).

Reviewer 2 Report

Comments and Suggestions for Authors

The manuscript ‘Rare prenyllipids in wild St. John's wort during three harvest seasons’ brings to our attention a study that examined the content of tocochromanols in leaves, flowers and buds of H. perforatum, collected in Poland in the years 2022-2024.
The authors have described all parts well but some of them need to be improved and/or modified according to the comments in the attached pdf file.

Author Response

We sincerely thank you for all the comments, remarks, and suggestions that have contributed to enhancing the manuscript and its scientific quality. The manuscript and supplementary materials have been improved accordingly. Provided changes are marked in red font. For literature we used references manager software therefore changes are not highlighted.

Reviewer 2

The manuscript ‘Rare prenyllipids in wild St. John's wort during three harvest seasons’ brings to our attention a study that examined the content of tocochromanols in leaves, flowers and buds of H. perforatum, collected in Poland in the years 2022-2024.
The authors have described all parts well but some of them need to be improved and/or modified according to the comments in the attached pdf file.

Comments 1: Please, insert references.

Response 1: Thank you for the comment. The references have been added—specifically moved closer to the cited values.

Comments 2: I think this part should be included more in the conclusions than in the results, as it is not a result explicitly obtained from this work. Similarly to the previous sentence, this part should not be included in this section. As with the previous sentences, these various references are more part of an introduction than a discussion, since they do not appear to be directly related to the results of the study conducted. They can be rephrased; they could be included in the conclusions.

Response 2: Thank you for the comment. We large partly agree with the reviewer's opinion; however, we also see added value in the discussion. We believe that we found some compromise. We have created an additional subparagraph “2.3. The potential applications, challenges, and future directions for tocotrienol-rich extracts of H. perforatum” and restructured the sentences to better align with the discussion. (Page 9, whole page, Page 10, top part).

Comments 3: Please enter the geographic coordinates of the individual sampling locations.

Response 3: Thank you for the comment. The geographic coordinates of the individual sampling locations have been added. (Page 10, top part).

Comments 4: Indicate who identified the plant species and where the voucher specimens were deposited.

Response 4: Thank you for the comment. The following information has been added: „Wild H. perforatum was identified according to a taxonomic guide https://powo.science.kew.org by Dr. Paweł Górnaś (PhD), who was previously trained by several botanists prior to the collection of the plants”. (Page 10, top part).

We would very much like to provide voucher specimen number(s); however, we have not deposited the Hypericum samples. The entire plant has been ground, making it impossible for us to obtain voucher specimen number(s). The plant was identified according to standard guidelines, and we have provided the most accurate description possible in the previous version of the manuscript, including photos of the H. perforatum and its aerial parts in the Supplementary Materials.

Comments 5: Please clarify the entire sentence as the insertion of the symbol ‘-’ may distort understanding of the data in some parts (it is not clear whether the symbol is used for sentence separation or as a negative sign).

Response 5: Thank you for the comment. This section of the methodology has been revised to eliminate any doubts. (Page 11, middle part).

Comments 6: Please set as superscript.

Response 6: Thank you for the comment. The revision has been incorporated into the manuscript.

Comments 7: The conclusions appear to be too compact. Extend the section and insert the sentences that were wrongly inserted in the other sections.

Response 7: Thank you for the comment. The conclusions part has been extended and supplemented: “The results of our study indicate that differences in the tocochromanols content between various parts of the plant and their stability under variable environmental conditions could form the basis for developing more sustainable cultivation practices for H. perforatum. Precisely determining the optimal harvest time and utilizing selected plant parts allows for the optimization of biologically active compound production while simultaneously minimizing waste and environmental impact. H. perforatum could thus serve as a model example of a medicinal plant whose cultivation contributes to biodiversity conservation and the implementation of sustainable agricultural practices. The findings may also support the development of cultivation systems for medicinal plants aimed at improving the predictability of both the quality and quantity of obtained raw materials, which is crucial for the pharmaceutical and cosmetic industries. Future research should investigate the genetic basis of tocotrienol accumulation in H. perforatum to identify genetic and/or biotechnological enhancement opportunities. The next step would be to develop sustainable and economically viable extraction and purification techniques aimed at obtaining high-quality products rich in health-promoting bioactive compounds. Additionally, exploring the synergistic effects of tocotrienols with other bioactive phytochemicals in St. John’s wort, e.g. hypericin and hyperforin, could lead to combined therapeutic applications. Finally, clinical studies are necessary to validate the pharmacological efficacy of the extracted bioactive molecules, particularly δ-T3.” (Page 12, bottom part, Page 13, top part).